# Integration of metagenome-assembled genomes with clinical isolates expands the genomic landscape of gut-associated *Klebsiella pneumoniae*

Samriddhi Gupta & Alexandre Almeida ✉

*Klebsiella pneumoniae* is an opportunistic pathogen causing diseases ranging from gastrointestinal disorders to severe liver abscesses. While clinical isolates of *K. pneumoniae* have been extensively studied, less is known about asymptomatic variants colonizing the human gut across diverse populations. Developments in genome-resolved metagenomics have offered unprecedented access to metagenome-assembled genomes (MAGs), expanding the known bacterial diversity within the gut microbiome. Here we analysed 656 human gut-derived *K. pneumoniae* genomes (317 MAGs, 339 isolates) from 29 countries to investigate the population structure and genomic landscape of gut-associated lineages. Over 60% of MAGs were found to belong to new sequence types, highlighting a large uncharacterized diversity of *K. pneumoniae* missing among sequenced clinical isolates. In particular, integrating MAGs nearly doubled gut-associated *K. pneumoniae* phylogenetic diversity, and uncovered 86 MAGs with >0.5% genomic distance compared to 20,792 *Klebsiella* isolate genomes from various sources. Pan-genome analyses identified 214 genes exclusively detected among MAGs, with 107 predicted to encode putative virulence factors. Notably, combining MAGs and isolates revealed genomic signatures linked to health and disease and more accurately classified disease and carriage states compared to isolates alone. These findings showcase the value of metagenomics to understand pathogen evolution and diversity with implications for public health surveillance strategies.

The species *Klebsiella pneumoniae* is a Gram-negative, facultative anaerobic opportunistic pathogen found in the human upper respiratory and intestinal tracts[1]. Historically, *K. pneumoniae* strains were classified based on their capsular types (K-types) and lipopolysaccharide O-antigen structures. However, the use of genetic markers and the wider adoption of whole genome sequencing led to an improvement in the characterization of *K. pneumoniae* diversity. Diancourt et al.[2] introduced a multi-locus sequencing typing (MLST) scheme based on seven housekeeping genes, enabling the

categorisation of *K. pneumoniae* populations into distinct sequence types (STs). These STs have been instrumental in understanding the genotypic and phenotypic diversity of *K. pneumoniae* subspecies, especially in the context of their virulence potential. Based on the presence of specific virulence genes, *K. pneumoniae* can be divided into classical (cKP) and hypervirulent (hvKP) strains. A previous study[3] identified five genotypic markers that effectively differentiate hvKP from cKP strains: *iucA* (aerobactin siderophore biosynthesis), *iroB* (salmochelin siderophore biosynthesis), *peg-344* (putative transporter)

Department of Veterinary Medicine, University of Cambridge, Cambridge, UK. ✉e-mail: aa2369@cam.ac.uk

and *rmpA/rmpA2* (regulators of capsule production). cKP strains are also more commonly associated with two major antimicrobial resistance (AMR) mechanisms: the production of extended beta-lactamases (ESBLs) and carbapenemases. These resistant strains have globally disseminated and, in 2019, *K. pneumoniae* was identified as one of the six pathogens responsible for over 250,000 deaths associated with AMR[4]. Consequently, the World Health Organization has classified carbapenem-resistant *Enterobacteriaceae*, including *K. pneumoniae*, as priority pathogens in urgent need of new treatments.

While significant advances have been made through genomic studies of *K. pneumoniae* clinical isolates, studies on human carriage strains circulating asymptomatically in the global population are limited. Although *K. pneumoniae* is part of the human gut microbiome, carriage strains may acquire pathogenic traits that further increase the risk of infection[5,6], especially among immunocompromised individuals[7]. *K. pneumoniae* strains colonising the gut microbiome have also been linked to gastrointestinal disorders such as inflammatory bowel disease[8], colorectal cancer[9] and diarrhoea[10]. Furthermore, it has been suggested that the intestinal tract serves as a major reservoir for transmission to other sterile sites[11] and increases the risk of extra-intestinal infections such as urinary tract infections (UTIs), bacteraemia, liver abscesses, meningitis, endophthalmitis and osteomyelitis. For instance, a prior study on a cohort of 498 patients found that *K. pneumoniae* gut colonization on admission was significantly associated with subsequent extraintestinal infections such as pneumonia, wound infections, UTIs and bacteraemia with sepsis[12]. Altogether, these studies demonstrate that gut colonization is a major risk factor for nosocomial *K. pneumoniae* infections, highlighting the need for research efforts investigating the pathogenic potential of gut-associated lineages.

Developments in metagenomic methods have transformed our understanding of the diversity of the species, strains and genes found within the human gut microbiome[13–16]. In metagenomics, the entirety of the genomic content of a sample is purified, fragmented and sequenced. The resulting DNA segments can then be assembled into contigs to reconstruct the genomes of the microorganisms present in the sample, which are referred to as metagenome-assembled genomes (MAGs)[17]. Large sequence catalogues derived from the human gut microbiome are now available in public databases such as the Unified Human Gastrointestinal Genome (UHGG)[18] catalogue. These include genomes recovered from faecal samples of various host states and geographic locations, providing access to diverse collections of microbial lineages naturally found in the human population.

In this study, we compiled 656 high-quality MAGs and isolates of gut-derived *K. pneumoniae* from 29 countries to explore their diversity and genomic properties. Our results highlight the value of integrating MAGs in pathogen genomic studies to obtain a more comprehensive understanding of the genomic diversity and infection risk of human gut-colonized lineages.

## Results

### Genotyping of gut-associated *K. pneumoniae*

To investigate the genetic features of *K. pneumoniae* lineages found in the human gut, we first compiled all high-quality MAGs (*n* = 317) and isolate genomes (*n* = 339) from the Unified Human Gastrointestinal Genome (UHGG) catalogue (Supplementary Data 1). The UHGG represents a comprehensive collection of isolates from two human gut culture collections[19,20] and public repositories (IMG[21], NCBI[22] and PATRIC[23]), alongside MAGs derived from >11,000 metagenomic samples worldwide. We combined both MAGs and isolates to understand the relative contribution of metagenomics in capturing the diversity of *K. pneumoniae* in the gut. We collected and curated additional genome metadata regarding the health status and country of origin from their associated samples. Of the 656 genomes, health status information was obtained for 521 genomes, with 132 being classified as carriage- (49

isolates, 83 MAGs) and 389 as disease-associated genomes (245 isolates, 144 MAGs). With regards to geographical distribution, the *K. pneumoniae* genomes spanned 29 countries, with the majority originating from China (*n* = 205) and the United States (*n* = 166). Overall, 505 genomes had complete metadata for both health status and country of origin (Fig. 1a).

We first assessed the distribution of sequence types (STs) derived from Kleborate[24] according to genome type (MAG or isolate). A total of 269 STs were identified, the majority of which (*n* = 168, 63%) were exclusively detected among MAGs, even after subsetting the analysis to countries with both MAGs and isolates (Fig. 1b). The most frequent STs detected in MAGs belonged to ST29, ST23 and ST65 (Fig. 1c), the latter of which was not represented by any isolate genome here included. However, these only represented 7–9 MAGs each, as there was a wide distribution of STs genotyped across all MAGs. Furthermore, 61.7% of MAGs belonged to new STs, which means they had at least 1 locus variant to a known ST (Fig. 1d). Notably, we observed that those more distantly related (> 2 locus variants) were primarily sampled from China and Fiji (Fig. 1d), emphasizing that these countries likely harbour more distinct *K. pneumoniae* lineages. Given that MAGs can often represent population genomes (that is, an amalgamation of multiple strains from the same sample) these results were further validated and confirmed after filtering for MAGs with low levels of strain heterogeneity (Supplementary Fig. 1). In the case of isolates, 143 genomes (42%) were assigned to three *K. pneumoniae* STs: ST11, ST258 or ST512, reflecting a more biased sampling towards lineages of clinical interest typically associated with *K. pneumoniae* carbapenemases (KPCs)[25,26]. Overall, our results highlight that genomes derived from metagenomic data are able to capture a more comprehensive, hidden diversity of *K. pneumoniae* that is missing from current gut isolate collections.

### Pan-genome patterns reveal unique MAG signatures

To explore the core and accessory genome of the collection of gut-derived *K. pneumoniae*, we used the pan-genome analysis tool Panaroo[27]. Given the use of MAGs can affect downstream pan-genome reconstructions, we tested Panaroo with various configurations to evaluate how robust the results were to parameter choice (Supplementary Fig. 2). Across different parameters, we obtained a mean pan-genome size of 21,160 genes (interquartile range, IQR = 20,738–21,559) and 4117 core genes (IQR = 4050–4182). Core- and pan-genome estimates were generally consistent across different settings, with maximum variations in pan-genome size and number of core genes of 5% and 3%, respectively. Downstream analysis were performed using the moderate filtering mode, 90% identity and non-merged paralogs to increase overall recall and mitigate the risk of technical artefacts leading to multiple variants of the same gene.

Using the above criteria, we then characterized pan-genome patterns of *K. pneumoniae* based on gene frequency and pan-genome accumulation curves (Supplementary Fig. 3). We observed a clear separation of core and accessory genes using a 90% threshold of gene presence (Supplementary Fig. 3a). Although the pan-genome curve using all accessory genes suggested an open pan-genome for *K. pneumoniae*, this was not the case when only considering genes found in at least >1% of the genomes (Supplementary Fig. 3b). This suggests that newly acquired genes are very rare within the whole *K. pneumoniae* population. Similarly, a 90% threshold for defining core genes revealed a more consistent trend compared to using a more strict 100% cut-off (Supplementary Fig. 3b).

Annotation of the pan-genome highlighted functional differences between the core and accessory genomes (Supplementary Fig. 3c). Accessory genes were significantly overrepresented (two-sided Fisher's exact test, *P* < 0.05) in functions related with replication,

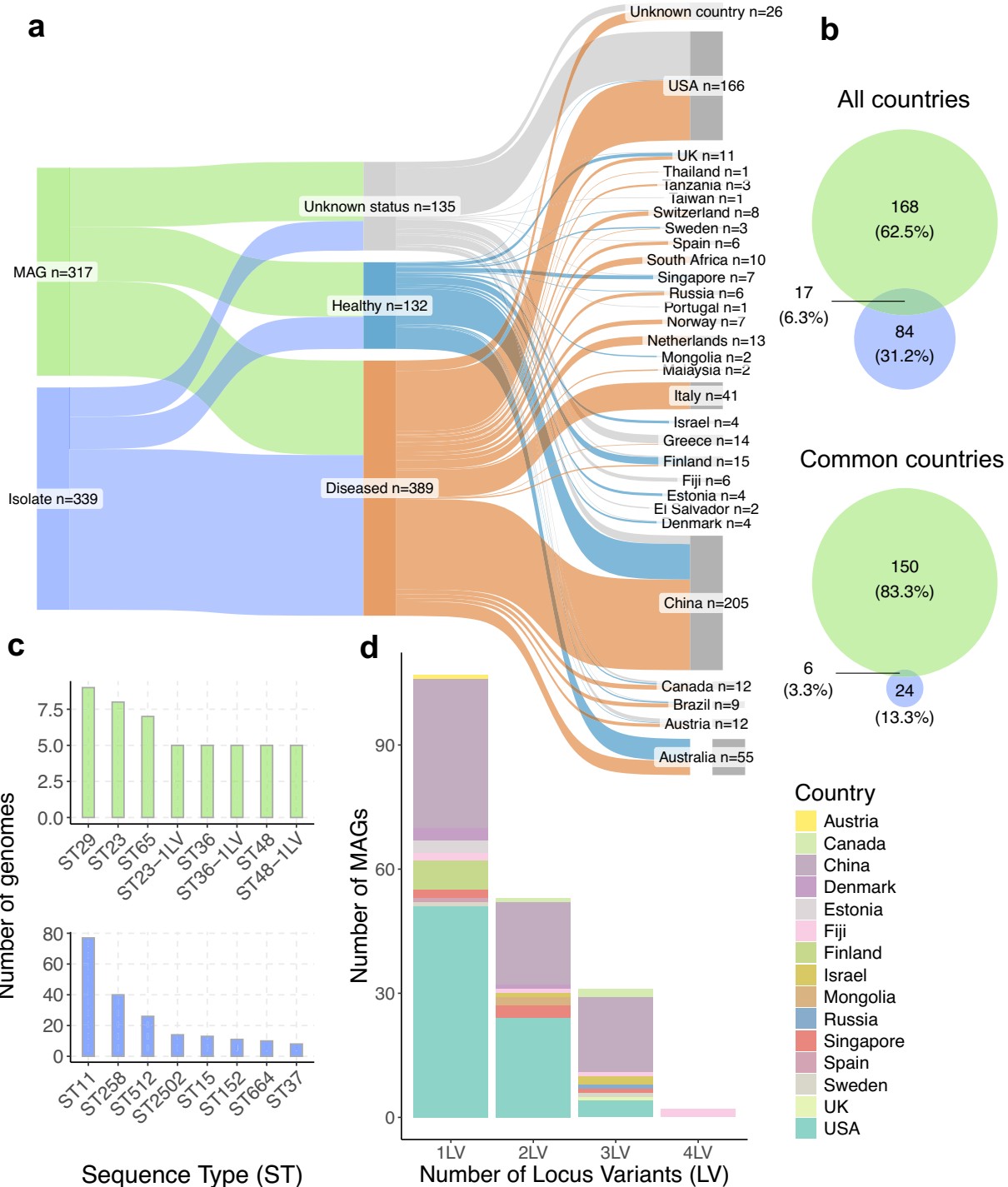

**Fig. 1 | Global collection of *K. pneumoniae* MAGs and isolates. a** Metadata distribution of the 656 *K. pneumoniae* genomes analysed. Data is partitioned into three metadata factors: genome type (metagenome-assembled genomes, MAGs or isolates), health status (diseased, healthy or unknown) and country of origin. **b** Venn diagram showing the intersection between sequence types (STs) detected among MAGs (green) and isolates (blue) across all countries (top) or only considering countries where both MAGs and isolate genomes were recovered (bottom). **c** Most prevalent STs of the gut-derived *K. pneumoniae* among MAGs (top) and isolates (bottom). **d** Distribution of MAGs detected per country based on the number of locus variants (mutations) identified in the MLST genes in relation to a known ST profile.

recombination and repair, as well as defence mechanisms. In contrast, core genes were predominantly associated with inorganic ion and amino acid metabolism, in addition to energy production. Of note, 61% of the accessory and 23% of the core genome of *K. pneumoniae* could not be assigned to a known functional category, highlighting the

extent of uncharacterized genetic diversity still to be explored even in well-known species of clinical relevance.

To further understand the unique genetic diversity of *K. pneumoniae* captured by metagenomics, we performed a dedicated analysis of genes found exclusively among MAGs within this genome

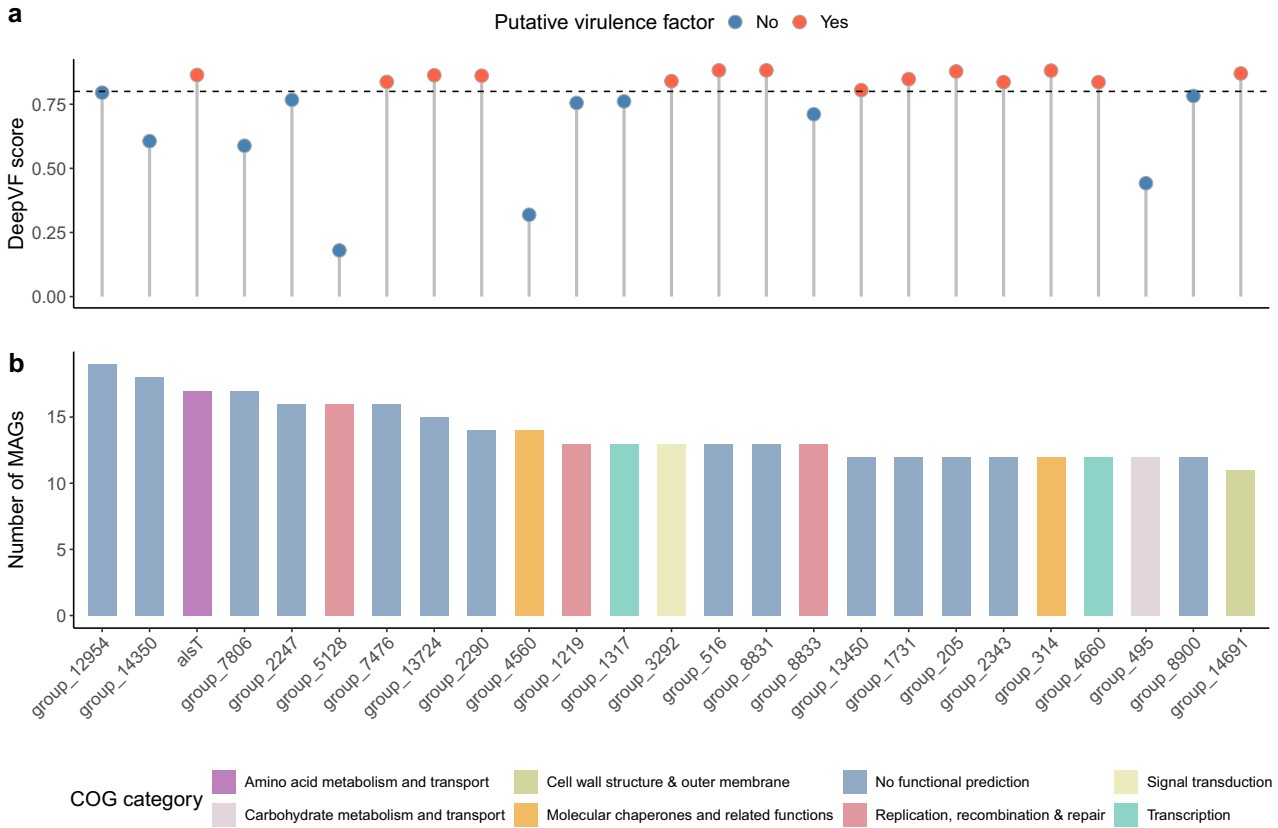

**Fig. 2 | Distribution and functional annotation of MAG-exclusive genes.**
**a** Probability scores obtained with DeepVF for each of the top 25 most prevalent genes exclusive to *K. pneumoniae* metagenome-assembled genomes (MAGs) from the human gut. A threshold of 0.8 (horizontal dashed line) was used to classify putative virulence factors. **b** Functional annotation based on Clustered Ortholo-gous Groups (COG) for each of the top 25 most prevalent genes detected solely among MAGs. Genes are ordered based on prevalence, represented by the number of MAGs in which a gene was detected.

collection (that is, missing from the *K. pneumoniae* gut isolate gen-omes). Seeing that MAGs are more prone to contamination from for-eign DNA, we only considered genes that were detected in at least 1% of genomes (> 6 genomes). This revealed 214 genes detected solely among MAGs, with a prevalence ranging from 7 to 17 MAGs (Fig. 2 and Supplementary Data 2) Those with known functions were most fre-quently involved in DNA replication, recombination and repair; inor-ganic ion transport and metabolism; and transcription regulation. However, the majority ($n = 129$) could not be assigned to a known function. We further searched the entire RefSeq protein collection of *K. pneumoniae*, which revealed that 36 of these MAG genes (17%) did not match known *K. pneumoniae* protein sequences. To further expand our search, we employed two deep learning approaches to screen for the presence of antimicrobial resistance and virulence-associated genes, using DeepARG[28] and DeepVF[29], respectively. Using a probability threshold of 0.8, 107 of the 214 MAG genes were flagged as putative virulence factors (Fig. 2a) and one gene was predicted to have antimicrobial resistance activity. The latter was assigned to a gene coding for a DNA-binding protein that did not match any known antimicrobial resistance (AMR) genes based on sequence similarity alone using the NCBI AMRFinderPlus[30] tool. Altogether, these results suggest that the *K. pneumoniae* MAGs here included harbour a unique genetic diversity that may be of clinical interest.

## Metagenome-assembled genomes expand *K. pneumoniae* diversity

Using the core genes identified with Panaroo, we investigated the phylogenetic relationship of the *K. pneumoniae* MAGs and isolate genomes here analysed. The phylogenetic tree built from the core

gene alignment exhibited a well-defined structure consistent with the predicted ST, with MAGs and isolates largely interspersed across the phylogeny (Fig. 3a). Considering the mixture of MAGs and isolate genomes could lead to increased technical variation, we compared the inferred population structure using different analysis approaches (Supplementary Fig. 3). We compared three clustering methods: i) the gene-based core tree obtained with Panaroo; ii) a SNP-based core tree generated by Snippy[31] with recombination removed; and iii) gene presence/absence patterns derived from the pan-genome data. There was a strong correlation between the three approaches tested (Mantel test, Pearson's r = 0.77–0.91; $P < 0.0001$), showing that the clustering structure was consistent across core genes, SNPs and the whole pan-genome (Supplementary Fig. 4). We then assessed the relationship between health state, geographical location, genome type (MAG or isolate) and source (community or hospital) with the afore-mentioned three clustering measures (Fig. 3b). Statistical analysis using a PERMANOVA test revealed that the specific health state (dis-ease name or condition) and country of origin had the strongest association (highest effect size, $R^2$) with the population structure of *K. pneumoniae* (Fig. 3b and Supplementary Table 1).

We further quantified how much genetic diversity within the gut-derived *K. pneumoniae* population was represented by each genome type. We performed a phylogenetic diversity (PD) analysis using Faith's PD method, which showed that the inclusion of MAGs pro-vided a nearly 2-fold (1.98) increase in phylogenetic diversity. This represented a 88% and 96% diversity increase among disease and carriage lineages, respectively, showing that MAGs are particularly useful at capturing a wider range of genomes from asymptomatic carriage. Of note, we did not identify any MAGs belonging to the

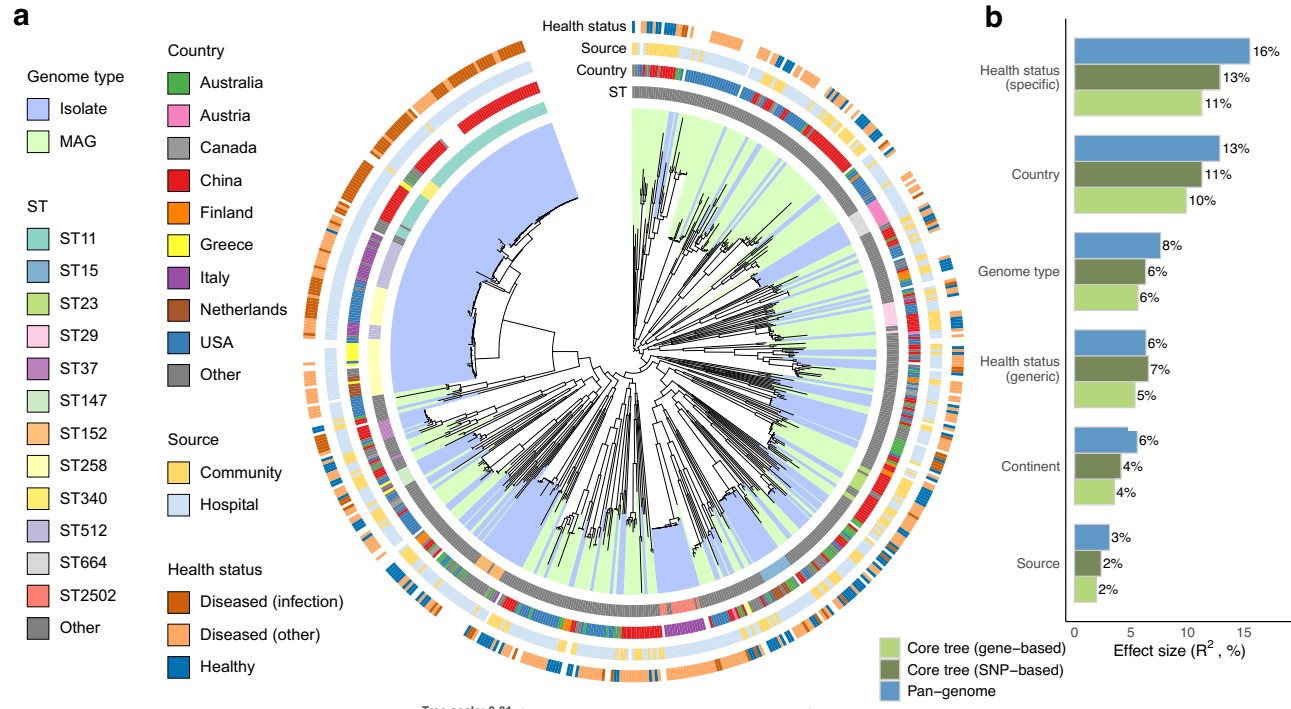

**Fig. 3 | Phylogenetic diversity of *K. pneumoniae* MAGs and isolates.**
**a** Phylogenetic relationship of the dataset of 656 gut-derived *K. pneumoniae* genomes based on phylogenetic distances of the gene-based core tree obtained with Panaroo. **b** Effect of various metadata variables on the population structure of *K. pneumoniae*, measured by the pairwise cophenetic distances using either the core trees (gene- or SNP-based) or the pan-genome distances (Jaccard). A PERMANOVA test was used to quantify the effect size ($R^2$) and statistical significance. All associations were found to be statistically significant ($P < 0.001$).

disease-associated clade dominated by the lineages ST11/ST258 (Fig. 3). Our collection of 317 *K. pneumoniae* MAGs is the result of screening >11,000 faecal metagenomic samples, the majority of which did not yield any high-quality *K. pneumoniae* MAGs. Therefore, the absence of the ST11 clade among our MAG dataset suggests it is likely very rare in the general population.

To further understand whether the MAGs here included matched *K. pneumoniae* isolate genomes from other sources, we conducted a comparative genomic analysis using Mash against all isolate genomes from the *Klebsiella* genus available on the NCBI RefSeq database ($n = 20,792$ genomes). Comparison of the 317 gut-derived *K. pneumoniae* MAGs against the RefSeq genomes showed a median genomic distance of 0.0034 (interquartile range, IQR = 0.0024–0.0052) with their closest matches (Fig. 4a and Supplementary Data 1). To evaluate whether Mash genomic distances correlated with the ST variation, we compared Mash genomic distances between genomes with different numbers of ST locus variants (LV). We found that genomic distances were significantly higher in genomes with greater LV differences (Supplementary Fig. 5; Wilcoxon rank-sum test, adjusted $P < 0.05$). This indicates that the number of LVs is generally associated with genomic divergence, rather than representing isolated SNPs.

Interestingly, the Mash genomic distances exhibited a bimodal distribution separated between a distance threshold of 0.005 (approximately equivalent to 99.5% average nucleotide identity, ANI), potentially reflecting a strain-level boundary. In fact, a previous study showed a pronounced intraspecies gap at 99.5% ANI among well-sampled bacterial species, suggesting the existence of defined intraspecies units within this ANI boundary[32]. We also observed that genomes with >2 LV had a median genomic distance >0.005, again supporting that they represent more divergent lineages from their ancestor ST (Supplementary Fig. 5). We detected 86 MAGs in particular with Mash genomic distances greater than 0.005 to their closest RefSeq sequence, highlighting understudied lineages without a

matching reference isolate genome. However, the remaining 231 MAGs did match a *Klebsiella* isolate genome from RefSeq, showing that these MAGs may be present in isolates from other body sites, hosts or environments outside the human gut. Phylogenetic analysis showed the new lineages overall conferred a 22% increase in phylogenetic diversity of gut-associated *K. pneumoniae*. These MAGs spanned distinct health states and geographical regions (Fig. 4b), but when considering countries where both isolates and MAGs were recovered, novel lineages provided the highest expansion in Singapore and the USA, with over a 6-fold phylogenetic increase in each country (Fig. 4c). Collectively, these results illustrate the value of using genome-resolved metagenomics for uncovering a more complete view of pathogen diversity and evolution worldwide. In particular, having representative genomes from these novel gut-adapted lineages provides reference sequences that will facilitate future efforts to track their dissemination and evolution in both health and disease.

**Genomic signatures of carriage and disease**
Having a collection of MAGs and isolates across diverse health states provided the opportunity to investigate genetic features associated with carriage and disease. We performed a microbial Genome Wide-Association Study (mGWAS) at both gene and SNP level. Because of inherent genomic differences between MAGs and isolate genomes, we included 'genome type' as a covariate in the models, and further excluded any genes found to be significantly associated with either MAGs or isolates (see Methods for further details). In the end, we identified 458 unique genes differentially present according to health status (FDR < 5%, Supplementary Data 3): 339 associated with infection or carriage (Fig. 5 and Supplementary Fig. 6), and 152 from the analysis of all disease lineages. At the SNP level, only one mutation (CP012745:g.1749711 C > T) was found to be significantly over-represented among genomes from infection. This SNP was located in an intergenic region between two glutathione S-transferase genes.

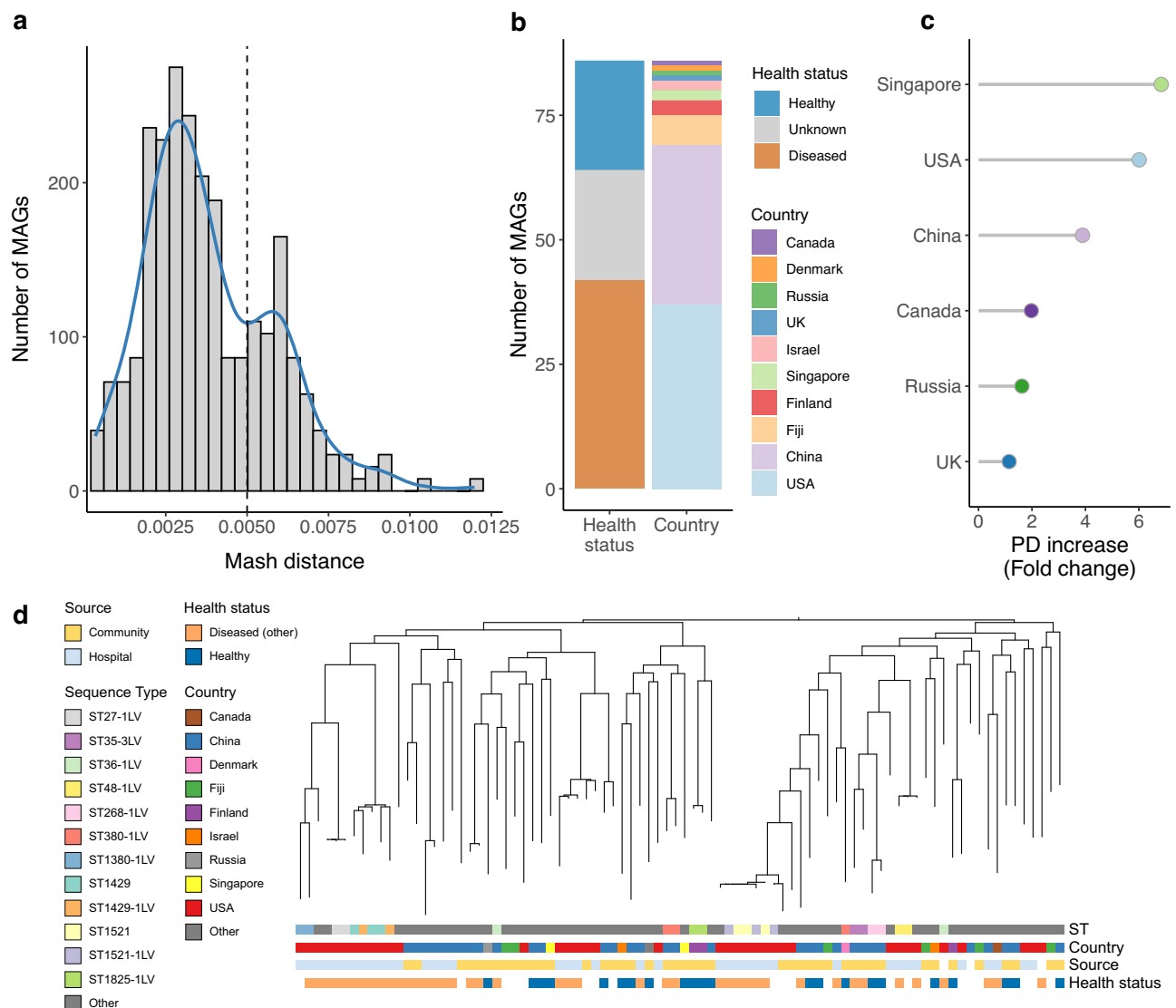

**Fig. 4 | Comparison of metagenome-assembled genomes with reference isolate genomes. a** Distribution of genomic distances calculated with Mash between the *K. pneumoniae* metagenome-assembled genomes (MAGs) and their closest reference genomes when compared against 20,792 *Klebsiella* isolate genomes from the NCBI RefSeq database. The histogram shows the frequency of MAGs at different Mash distance intervals, with a density curve (blue line) overlaid to illustrate the overall distribution. A threshold of 0.005 (vertical dashed line) was used to define lineages without a close reference genome. **b** Metadata distribution of the 86 MAGs representing lineages without a reference isolate genome available on NCBI RefSeq. **c** Phylogenetic diversity (PD) increase provided by the 86 MAGs without a matching isolate, compared to using isolates alone. Only countries where both MAGs and isolates were obtained were evaluated for this analysis. **d** Phylogenetic tree of the 86 MAGs with a genomic distance >0.005 in relation to NCBI RefSeq, annotated based on their ST, country, source (community or hospital) and health status.

Glutathione S-transferases are a family of enzymes involved in the detoxification of endogenous and xenobiotic compounds[33].

At the gene level, we focused on the 339 candidate genes specifically associated with infection or carriage, as these are potentially more directly linked to the pathogenesis of *K. pneumoniae*. Grouping the 339 significant genes into Clusters of Orthologous Groups (COGs) showed that categories involved in transcription and amino acid metabolism were overrepresented in carriage (two-sided Fisher's exact test, FDR < 5%), while DNA replication and repair were more predominant in disease (at an FDR = 12%, Fig. 5b). However, 207 genes could not be assigned to a known function, representing a large genetic repertoire that may underly uncharacterized disease mechanisms. Among the most significant carriage genes with a predicted function was a genetic cluster containing a transcriptional repressor similar to the Ferric uptake regulator (Fur) family. In contrast, infection-associated genes with the highest effect size were related with restriction modification systems, phage proteins and

polysaccharide biosynthesis. Given geographical biases in our dataset, we repeated the GWAS by considering only countries where both health and infection genome data were available (Australia, Brazil, China, and the UK). Importantly, 85% (288/339) of the genes remained significant using this subset, and 70% (237/339) when further subsetting to isolate genomes only. This not only emphasizes the robustness of the observed signal, but also highlights how the use of MAGs was able to reveal additional signatures of health and disease.

To further assess how well the pan-genome diversity was able to distinguish carriage from infection-associated lineages, we trained supervised machine learning models (ridge regression, random forest and gradient boosting) to classify *K. pneumoniae* genomes based on the health status (infected or healthy) of their hosts. All models showed good classification performance (Area Under the Receiver Operating Characteristic Curve, AUROC > 0.9; Fig. 5c and Supplementary Fig. 7), even by broadening the analysis to all disease lineages (median AUROC > 0.8; Supplementary Fig. 7). Notably, models trained with

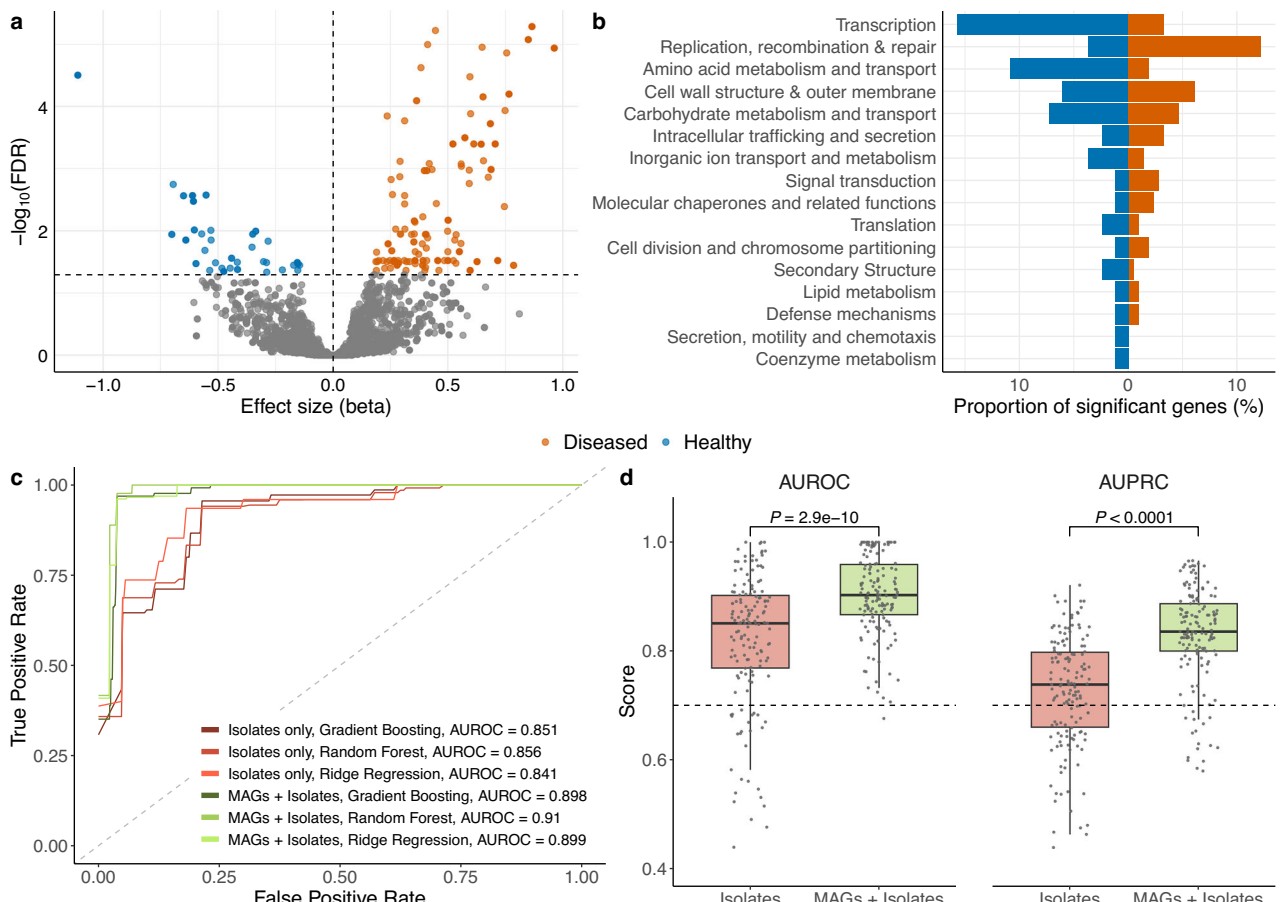

**Fig. 5 | Genomic signatures of *K. pneumoniae* in carriage and infection.**
**a** Distribution of effect sizes (beta) and -log₁₀–transformed adjusted *P* values (False Discovery Rate, FDR) of genes significantly present or absent among infection-associated *K. pneumoniae* genomes. **b** Proportion of the significant genes assigned to different Clustered Orthologous Groups (COG) categories. Orange bars to the right indicate the number of genes overrepresented in infection lineages, while the blue bars show genes overrepresented in carriage. **c** Receiver Operating Characteristic (ROC) curve of the machine learning results linking the pan-genome patterns with the health state of the individual sampled. **d** Comparison of Area Under the Receiver Operating Characteristic Curve (AUROC) and Area Under the Precision-Recall Curve (AUPRC) scores of machine learning (ML) models classifying carriage and infection lineages using isolates alone or MAGs together with isolate genomes. Each box represents the interquartile range (IQR) across 150 independent seeds (*n* = 50 per model). The centre line within the box represents the median score. Whiskers are shown extending to the furthest point within 1.5 times the IQR from the box. *P* values were derived from a two-sided Wilcoxon rank-sum test (AUROC: *P* = 2.9 × 10⁻¹⁰; AUPRC: *P* < 0.0001).

both MAGs and isolates performed significantly better (two-sided Wilcoxon rank-sum test, *P* < 0.001) than those with isolates alone, especially when evaluating model performance with the Area Under the Precision-Recall Curve (AUPRC, Fig. 5d). Therefore, these results show that combining both MAGs and isolates significantly improves the ability to distinguish genomic signatures related with health and disease.

## Discussion

Pathogen genomic studies have predominantly focused on the analysis of clinical isolates to derive biological insights into bacterial pathogenesis. In this study, we analysed over 600 metagenome-assembled genomes (MAGs) and isolate genomes of *K. pneumoniae* to improve our understanding of the evolution, phylogeography, and genetic features of gut-associated lineages. We focused on genomes from the human gut given the importance of gut colonization as a risk factor for *K. pneumoniae* infections[1,11,12,34]. Our findings expand on previous work investigating the global genomic diversity of *K. pneumoniae* in humans across different health states[35], including a prior phylogenetic analysis of hypervirulent strains found in the human gut[36].

Our study demonstrates that metagenomics is a powerful tool for uncovering the hidden diversity of *K. pneumoniae* in the gut microbiome. Notably, over 60% of MAGs differed from known STs, with 86 MAGs in particular exhibiting a genomic distance >0.5% to any known isolate. Interestingly, these new lineages expanded the *K. pneumoniae* gut diversity more substantially among countries with large population sizes (China or USA) or that are underrepresented among isolate collections (Singapore and Fiji). MAGs also harboured more than 200 unique genes, many of which were associated with either virulence or antimicrobial resistance. These results underscore the importance of using metagenomics to identify previously undetected genes and lineages that may impact *K. pneumoniae* pathogenicity or resistance.

A further mGWAS analysis integrating MAGs and isolates to compare health- and disease-associated lineages revealed 458 genes that were significantly different between health states. These included genes related with the Fur transcriptional regulator, surface polysaccharides and restriction modification systems. Members of the Fur family have been shown to repress siderophore activity which in turn can reduce *K. pneumoniae* virulence[35,37]. An overrepresentation of these genes among carriage strains suggests they may attenuate the pathogenic potential of *K. pneumoniae*. Surface polysaccharides, which were found to be overrepresented in infection-associated genomes, have been previously implicated in *K. pneumoniae*

pathogenesis[35]. However, the role of restriction modification genes and phages in disease has been less studied. Restriction modification systems are responsible for bacterial protection against foreign DNA, such as that encoded by bacteriophages[38]. We hypothesize that the ability of these lineages to resist infection by certain bacteriophages may provide a fitness advantage compared to carriage strains.

Although we reveal important insights into the genomic features of gut-associated *K. pneumoniae*, it is worth noting some of our study's limitations. Despite our strict filtering criteria, MAGs should still be treated with caution compared to isolate genomes due to a higher risk of contamination from closely-related species[39]. Furthermore, even though we analysed a globally diverse genome collection, our samples were overrepresented in certain well-studied regions such as China and the United States, with some countries only represented by either carriage or disease genomes. Lastly, given the genomic focus of our study, the uncharacterized candidate genes here identified will inherently require further experimental testing to explore their mechanistic role in the aetiology of disease.

In summary, our work underscores the importance of integrating MAGs with clinical isolates for population-based genomic analyses, identifying specific features and lineages of *K. pneumoniae* that are missed by relying solely on the analysis of isolate genomes. Ultimately, improving our understanding of the diversity of *K. pneumoniae* colonizing the human gut could lead to the development of targeted public health measures and therapeutic approaches aimed at reducing infection risk.

## Methods

### Genome collection and quality control
We first extracted all *Klebsiella pneumoniae* genomes derived from faecal samples available in the Unified Human Gastrointestinal Genome (UHGG)[18] v.1.0 database (*n* = 985). No statistical method was used to predetermine sample size. This dataset comprised two genome types: metagenome-assembled genomes (MAGs) and isolates. Isolate genomes within the UHGG were originally retrieved by surveying the IMG[21], NCBI[22] and PATRIC[23] databases for genome sequences annotated as having been isolated from the human gastrointestinal tract. This set was complemented with bacterial genomes belonging to two human gut culture collections: the HBC[19] and CGR[20]. With regards to MAGs, those included in the UHGG were obtained from the studies of Pasolli et al.[13], Almeida et al.[15] and Nayfach et al.[14]. Given MAGs are generally of lower quality than isolate genomes, we performed strict quality control procedures. First, we applied a filter of >90% completeness and <5% contamination based on the genome statistics obtained with CheckM[40] v.1.0.11. CheckM uses a database of single-copy marker genes to estimate genome completeness and contamination, looking specifically at the composition of the core genome. Thereafter, we used GUNC[41] v.1.0.3 to further exclude genomes with both a 'clade_separation_score' >0.45 and 'contamination_portion' >0.05. In contrast to CheckM, GUNC investigates the full complement of genes to detect and quantify genome contamination (that is, the incorrect placement of genes from other species within a genome). It leverages an entropy-based measure of lineage homogeneity across contigs to determine if they share a consistent taxonomic assignment. Lastly, all genomes underwent taxonomic verification for *K. pneumoniae* using Kleborate[24] v.2.4.1 using default parameters, and subsequently dereplicated to remove duplicate or nearly identical genomes based on a Mash[42] distance of 0 with a sketch size of 10000. The final filtered and dereplicated dataset comprised a total of 656 genomes (317 MAGs and 339 isolate genomes; Supplementary Data 1). Strain heterogeneity values for each MAG were retrieved from the UHGG[18] catalog. These were originally calculated with the CMseq tool[13] by mapping the metagenomic reads from the sample used to generate the MAG. The level of strain heterogeneity was estimated by calculating the number of nonsynonymous substitutions detected out of all positions mapped with a depth of coverage of at least 10 reads and

base quality of at least 30 (a minimum of 100 positions were needed to estimate strain heterogeneity).

### Metadata curation and genotyping
Metadata regarding health status, country of origin and source for each genome was gathered from the National Center for Biotechnology Information (NCBI) or the European Nucleotide Archive (ENA). Furthermore, a review of associated research papers (if available) was conducted to supplement the metadata obtained. Carriage strains were defined as those obtained from individuals explicitly classified as healthy in their original study. Individuals classified as diseased were further analysed in two groups: (i) those with any disease, such as infections, colorectal cancer, autoimmune disorders, liver disease, among other conditions considered as risk factors for *K. pneumoniae* infection; and (ii) those specifically with conditions directly associated with or caused by the colonization of pathogenic *K. pneumoniae* strains, namely infections and diarrhoeal diseases. Sequence types (STs) for each genome were derived with Kleborate, which performs sequence alignment against an established MLST scheme[2] comprising seven loci (more information can be found in the associated BIGSdb resource: https://bigsdb.pasteur.fr/klebsiella/).

### Pan-genome analysis
We constructed a pan-genome of the 656 *K. pneumoniae* genomes selected using Panaroo[27] v.1.3.3, which enabled us to differentiate between the core and the accessory genes. We compared the use of different parameters to assess the impact of varying stringency levels in identifying the core genome. Specifically, Panaroo was run with 'mode' moderate and strict, and for each mode configuration we tested a sequence identity threshold of both 90% and 95%. Additionally, the approach of handling paralogous genes was compared between merging and keeping them separate. The core genome threshold was set to 90%.

Functional analysis of the core and accessory genomes was performed using eggNOG-mapper[43] v.2.1.3. Genes were classified based on their assigned Cluster of Orthologous Group (COG) functional category. The percentage of genes in each COG category was calculated separately for both core and accessory genomes.

Based on the pan-genome patterns, we further extracted gene clusters found exclusively among MAGs (that is, missing from any of the isolate genomes here included). As MAGs have a greater risk of containing foreign genes due to contamination introduced during the binning process, we used an additional prevalence filter of 1% (that is, genes that were detected in 6 or fewer genomes were excluded).

### Phylogenetic analyses
We used Snippy[31] v.4.6.0 (18) to extract single nucleotide polymorphisms (SNPs) of each *K. pneumoniae* genome against reference GUT_GENOME147590 (Supplementary Data 1), representing a finished isolate assembly. Gubbins[44] v.3.3.1 was utilized to identify and remove regions of recombination from the SNP alignment file and to construct a filtered phylogenetic tree. In parallel, a gene-based phylogenetic tree was also built using the approximate maximum-likelihood method implemented in FastTree[45] v.2.1.11, based on the core gene alignment generated by Panaroo. Lastly, a pan-genome tree was derived by estimating pairwise Jaccard distances of gene presence/absence patterns across all genomes. The SNP-based and pan-genome trees were visually compared with the gene-based core tree from Panaroo through a tanglegram using the 'tanglegram' function of the 'dendextend' R package[46]. To construct the tanglegram, trees were converted to ultrametric distances (function 'chronos'; 'ape' R package[47]) and rooted at their midpoints (function 'midpoint.root'; 'phylotools' R package[48]). These were subsequently converted into dendograms (function 'as.dendogram'; base R) and reordered to ensure that

corresponding taxa were positioned similarly in both trees. A statistical analysis using the Mantel test was conducted (function 'mantel'; 'vegan' R package[49]), using the Pearson correlation method with 999 permutations, to assess the correlation between the distances derived from each tree.

To investigate the association between the population structure of our dataset and key metadata factors, such as health status (diseased or healthy), genome type (MAGs or isolates) and country of origin of the samples), we performed a permutational multivariate analysis of variance (PERMANOVA) test using the 'adonis2' function from the 'vegan' R package[49]. As the 'adonis2' function requires a distance matrix as an input, the previously calculated cophenetic or Jaccard distances from the gene, SNP or pan-genome trees were used. Modelling was performed using the metadata factor of interest as the independent variable, and the distance matrix as the dependent variable.

To assess phylogenetic diversity (PD), we employed the Faith's PD method, which quantifies diversity as the sum of all branch lengths. The phylogenetic tree constructed using Panaroo was rooted at midpoint and filtered to create a subset of the tree with isolates only. The increase in phylogenetic diversity provided by the MAGs was calculated as: $(PD_{all} - PD_{isolates}) / PD_{isolates}$.

### Sequence comparison with public databases

To determine the similarity of gut-derived *K. pneumoniae* metagenome-assembled genomes (MAGs) to cultured *K. pneumoniae* genomes from other habitats, we performed a comparative genomic analysis using Mash[42] v.2.3 which calculates pairwise genome distances based on shared k-mer content. Each *K. pneumoniae* MAG was compared against a database of all cultured genomes from the *Klebsiella* genus available on NCBI RefSeq release 219 ($n = 20,792$ genomes). Thereafter, we identified the closest reference genome for each MAG based on the lowest Mash distance. A Mash distance threshold of 0.005 was used to determine which MAGs represented lineages without a reference isolate genome.

We performed a similar comparison of genes found exclusively among MAGs against the NCBI RefSeq protein collection of *K. pneumoniae*. After performing a 'blastp' search, we defined a positive match when >80% of the query MAG protein aligned with >90% amino acid identity against a RefSeq protein.

### Resistance and virulence gene analysis

Genes identified exclusively among *K. pneumoniae* MAGs were further annotated with eggNOG-mapper. In addition, we used AMRFinderPlus v.3.11.26[30], DeepARG v.1.0.4[28] and DeepVF (release 2019-08-30)[29] to screen for the presence of genes associated with either antimicrobial resistance (AMR) or virulence. AMRFinderPlus was run with option '--plus' using the database release 2023-04-17.1. This tool uses sequence similarity approaches to identify putative genes linked to AMR or virulence. In contrast, DeepARG and DeepVF use a combination of deep learning approaches trained on known antimicrobial resistance and virulence factors to identify both known and novel genes. With both tools, a probability threshold of 0.8 was used to define the presence of putative AMR and/or virulence factors among the MAG-exclusive genes.

### Microbial genome wide-association study (mGWAS)

Pyseer[50] v1.3.11 was used to identify genetic features (genes and single nucleotide polymorphisms, SNPs) associated with health status. The gene presence/absence matrix generated with Panaroo was used as input for Pyseer to identify genes significantly present or absent in disease-associated *K. pneumoniae* genomes. However, to account for MAG incompleteness and/or contamination, only genes present in >1% and <90% of the population were analysed. Parallel to this, SNP data derived from Snippy were used to identify mutations associated with either carriage or diseased genomes.

mGWAS was performed solely on the genomes whose health status and country of origin was available. To account for population structure, the Fast-LMM (Factored Spectrally Transformed Linear Mixed Models) algorithm was utilized with a similarity kinship matrix. This was achieved using the script 'phylogeny_distance.py' from Pyseer with the '--lmm' option to signify the use of a linear mixed model that considers the phylogenetic distance as a measure of genetic similarity. Country of origin and genome type (MAG or isolate) were additionally used as covariates in the mixed effect model. Statistical significance was determined after multiple testing correction using the Benjamini-Hochberg procedure and False Discovery Rate (FDR). An adjusted *P* value threshold <0.05 was used to find genes and SNPs significantly associated with health or disease states (Supplementary Data 3). As an additional control, a separate mGWAS analysis was performed to identify genes and variants associated with genome type (MAG/isolate), and these were excluded from the disease analysis.

To elucidate the biological implications of our mGWAS findings, significant genes were annotated via eggNOG-mapper. The output from eggNOG-emapper included detailed descriptions of each gene and an associated functional category based on the Clusters of Orthologous Groups (COGs) classification. Uncharacterized genes were considered those either without an eggNOG annotation, or with an 'S' (Function Unknown) COG classification.

### Machine learning classification

We trained three supervised machine learning (ML) models (ridge regression, random forest and gradient boosting) to distinguish disease- from carriage-associated *K. pneumoniae* genomes. ML models were trained based on the pan-genome presence/absence data. Features were pre-processed to only include those found between 1% and 90% frequency. ML models were run using a custom workflow adapted from the 'mikropml' R package[51,52] (https://github.com/alexmsalmeida/ml-microbiome). Only genomes whose metadata for both the health status and country of origin was available were utilised. Model training and hyperparameter tuning was performed on 80% of the data using a 5-fold cross-validation, while the other 20% were used for testing with the best hyperparameter setting. To evaluate model performance, each analysis was repeated with 50 unique random seeds. Additionally, the ST of the genomes was used as the grouping variable to account for population structure (that is, genomes from the same ST were kept together in either the training or test set). Summary statistics (mean, median, interquartile range) for the Area Under the Receiver Operating Characteristic Curve (AUROC) and the Area Under the Precision-Recall Curve (AUPRC) were used to evaluate model performance. Models trained with MAGs and isolates or isolates alone were compared to assess the diagnostic value of the MAGs for disease classification using a two-sided Wilcoxon rank-sum test. When performing model training with both MAGs and isolates, features found significantly associated with 'genome type' based on the mGWAS results were first excluded.

### Reporting summary

Further information on research design is available in the Nature Portfolio Reporting Summary linked to this article.

## Data availability

All genomes used are publicly available in the Unified Human Gastrointestinal Genome catalogue (https://ftp.ebi.ac.uk/pub/databases/metagenomics/mgnify_genomes/human-gut/v1.0/) and the European Nucleotide Archive (see Supplementary Data 1 for list of accession codes and genome identifiers). Pan-genome data files (gene presence/absence matrix and FASTA file) are available in: https://doi.org/10.6084/m9.figshare.27961089.v2.

## Code availability

Custom code and scripts are available in the following GitHub repository: https://github.com/microfundiv-lab/KpMAGs[53].

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

## Acknowledgements
The authors thank Ana Catarina da Silva for assisting in the curation of the sample metadata. We also thank Sebastian Bruchmann, Efrat Muller and all members of the Microbiome Function and Diversity group for helpful feedback and suggestions. Funding was provided by a Career Development Award from the Medical Research Council (MR/W016184/1) to A.A.

## Author contributions
S.G. performed the genomic analyses and wrote the manuscript draft. A.A. supervised the work, assisted in the analyses and edited the manuscript.

## Competing interests
The authors declare no competing interests.
