## [Transparent Peer Review file · Nature Communications]

Integration of metagenome-assembled genomes with clinical isolates expands the genomic landscape of gut-associated *Klebsiella pneumoniae*

Corresponding Author: Dr Alexandre Almeida

A version of this paper was originally rejected for publication by Nature Communications, however that decision was reconsidered after appeal by the authors.

Version 0:

Reviewer comments:

Reviewer #1

(Remarks to the Author)

The manuscript by Gupta et al. uses the publicly available data on metagenome-assembled genomes with clinical isolates of *Klebsiella pneumoniae* and aims to find the genomic signatures of this species in carriage and disease. I have several major concerns regarding the analysis methodology and interpretation of results. Further, in my opinion, the manuscript lacks in advancing our understanding in the area. Some of my suggestions are provided below

1. Details on the selection of 662 genomes (319+343) are confusing. While the authors do explain the origin of MAGs, the selection criteria of isolate genomes used is completely missing.
2. The lack of difference in virulence properties of carriage and disease genomes is reported. While authors also mention that disease-associated genomes in the gut can increase the risk of subsequent nosocomial infections. However, the lack of any difference in virulence properties in carriage vs. disease genomes suggests that infections from both groups are equally likely.
3. Authors should mention the carriage/disease genomes ratio in MAGs and isolate groups.
4. Was there any redundancy in the selected genomes? Did you use any dereplication method to cluster genomes and remove any redundancy?
5. Line 165-166: report p-values in all comparisons, even the non-significant ones.
6. Line 204: needs some revision. Also, the results based on analysis of a single species should not be extended to the whole metagenomic community and host health states.
7. Line 219 & 241: Please use consistent names for groups throughout the manuscript (disease/carriage, infection/carriage, carriage/disease state). This will improve the paper's readability.

(Remarks on code availability)

I have gone through the link and confirm that the code and other information is available at Github but I have not used it thus cannot comment on the usability or user friendliness of the code.

Reviewer #2

(Remarks to the Author)

In the study by Gupta et al, the authors collate a collection of 319 metagenome-assembled genomes and 343 isolate whole genomes from gut carriage/disease to examine the genome diversity of carriage versus disease-associated Kp. My concern is that this does not represent a systematic collection of genomes, and there will likely be different sampling biases underlying the data collection/sequencing of MAGs versus isolate whole genome data, which will likely impact the observations/conclusions that the authors have made. For example, most studies looking at clinical infections select for AMR. Additionally, it appears that the majority of MAGs and isolates are not linked/associated with the same host/geography (i.e. there is a disproportionate number of health and/or disease associate genomes within different geographies) - is there

any point then in trying to draw comparisons like ST trends and genome markers between carriage versus disease associated genomes when differences may be reflective of differences in the circulating clones/isolates within particular geographies?

Other Comments:

- Line 100: the authors state here that genomes were classified as carriage or disease associated - was the metadata stated as such as were there various categories that were collapsed down into these two categories? If so, are they authors able to elaborate here on these definitions?
- Lines 107-112: the authors state that there is more variability in the ST distribution of disease genomes versus carriage isolates but is this again due to differences in the sampling/geographies? For example, ST258 is a common carbapenem resistant Kp lineage in the US and ST512 in Italy, but there appears to be no genome data from 'healthy' patients from the US or Italy?
- Lines 112-113: Can the authors elaborate on how many genomes from the disease versus carriage were these 20 STs represented? Any comments on country distribution? i.e. Are genomes with the same ST/carriage/disease being detected in the same country?
- Lines 174-176: The authors state here that ST11/ST258 were absent from >11,000 faecal metagenomic samples; however the authors only looked at a subset of these samples (662 MAGs/isolates), so it's unclear how they can extend this observation (or lack thereof) to the entire sample collection? Additionally, will some of these 11,000 metagenome samples lack Kp - the authors mention n=985 Kp genomes from faecal samples in the methods section.
- Lines 196-197: The authors should briefly clarify here what the resistance and virulence scores measure - I've just seen that they provide the description of this in the methods, but should still give a brief explanation in the results, particularly if the methods comes after the results section.
- Lines 197-199: Can the authors provide some quantification here between the differences in the scores for the carriage versus disease cohorts (i.e. the mean or median)?
- Lines 198-238: Are some of these observations driven by Kp clone characteristics? For example, the authors state resistance scores in the disease cohort are significantly higher but there is a higher proportion of MDR Kp (e.g. ST11, ST258, ST512) in the disease cohort, which carry higher loads of AMR.
- Lines 446-449: Can the authors provide a little more explanation on the purpose for CheckM and GUNC? Is the >90% completeness only looking at core/chromosomal sequence?
- Line 525: Can the authors comment on why this threshold was selected?
- Figure 1: For panel B, it would be helpful to see a breakdown for each of the columns (i.e. ST counts) by MAGs versus isolate WGS
- Figure 4: Similarly for all panels, it would be helpful to see which of the genomes correspond to MAGs versus isolate WGS, and whether this influences assembly of plasmids, which often carry the virulence loci and AMR genes that are captured by the virulence/resistance scoring of Kleborate

Minor comments:

- Line 36: Gram should have a capital G
- Line 113 and 116: Typo, figure reference should be to Fig. 1C and not 2C
- Lines 228-238: a lot of this text reads like discussion text rather than results
- Line 255: do the authors mean >300 i.e. some of these 600 were whole genome sequenced from isolates?
- Line 265: unclear how this study would add to previous works looking at global diversity of Kp in animals as this study does not contain any sampling from animals

(Remarks on code availability)

Version 1:

Reviewer comments:

Reviewer #1

(Remarks to the Author)

The work does not provide any new significant information, or methodology or any significant finding in the area of metagenomics or disease biology

(Remarks on code availability)

I just checked its availability but did not examine it.

Reviewer #2

(Remarks to the Author)

The authors have addressed many of the reviewer comments, and the revised version is an improvement to the original manuscript. However, my concerns over the study's dataset remain - there are biases in the dataset which means any conclusions comparing diseased versus carriage will be flawed. As an example, the authors highlight less diversity amongst disease associated lineages and that these are dominated by ST11, ST258 and ST512; this is to be expected as it reflects the bias towards capturing prominent MDR clones that frequently cause outbreaks in clinical settings. The addition of MAGs does add value in expanding the availability of carriage associated genomes; are the authors able to reframe the narrative to

focus on this aspect rather than disease versus carriage?

(Remarks on code availability)

Version 2:

Reviewer comments:

Reviewer #2

(Remarks to the Author)

In the manuscript by Gupta et al, the authors use human gut metagenome-assembled (MAGs) and isolate genomes to highlight the value of using MAGs at capturing additional diversity within the gut that otherwise would be missed using isolate whole genome data alone. Some of these conclusions have been derived based on their MAGs having ST locus variant calls; I can see two potential issues with this whereby the approach doesn't provide enough resolution to confidently distinguish two isolates as divergent (i.e. a single SNP within any one of the 7 genes can result in a locus variant being called, and secondly, have the authors considered that some of these locus variant calls can arise simply due to the impact of sequencing errors associated with shallow sequencing depth that can impact MAG accuracy? In the last results section, the authors look for genomic signatures associated with carriage versus disease across their dataset of MAGs and isolate genomes. Are the authors able to comment on whether the same results would have been attained had they used isolate genome data alone? This would be one way to demonstrate impact/added value of including MAGs for analysis. Lastly, the authors mention that 25% of the MAGs were divergent but this means that 75% of the MAGs have a closely related RefSeq match (<0.005 mash distance); based on this perhaps the added value of adding MAGs to pathogen genomic studies is only incremental?

Other comments:

Line 26: missing among 'sequenced' clinical isolates

Line 92: In the introduction, the authors emphasize the utility of MAGs for gleaning insights into colonising Kp/the microbiome; can the authors clarify here why they've also included isolate genomes?

Lines 109-110: Did any of the isolate genomes correspond to these STs?

Lines 113-115: Here the authors use ST locus variants as a measure of phylogenetic diversity and to highlight China/Fiji as therefore harbouring more distinct lineages; I don't think using ST variation can provide enough resolution to make this claim. A single SNP can result in a locus variant being called. Have the authors checked their phylogenetic tree to confirm that these genomes with >2 locus variant calls are elsewhere in the tree?

Line 119: typo, should be 'STs' instead of strains. Authors should probably clarify that strains with these STs generally have/or are typically associated with KPCs. Were there any MAGs that had these STs?

Line 126: can the authors clarify why they've used Panaroo with different configurations?

Lines 154-157: I don't quite follow how the authors know that these genes are found exclusively in MAGs? i.e. which genomes/dataset have they compared their MAGs to?

Lines 189: The authors state here that the MAGs corresponded to lineages that are not represented by clinical isolates; how do they know this? Have the authors included clinical isolates in their analysis or is this just based on earlier observation of MAGs that do not have defined STs/matching to those that have been defined in clinical isolates? As highlighted earlier, STs do not provide enough resolution.

Lines 197-198: Wouldn't their 317 MAGs represent 317 faecal metagenomic samples rather than >11000 ? Or do the authors mean that no *K. pneumoniae* MAGs were assembled from the other 11000 metagenomes?

Lines 212-219: The authors highlight here unique MAGs but aside from these being novel lineages that otherwise haven't been captured by whole genome sequencing, they don't elaborate on what additional value this brings? Also, the majority (75%) of their MAGs fall under the 0.005 Mash distance (i.e. are closely related to a RefSeq genome?); but the authors have not made any comments on this result.

Lines 221-264: In this section,

Lines 282-283: This seems like quite a substantial number of unique genes associated with virulence/AMR - are these genes entirely absent from isolate whole genomes from NCBI etc?

(Remarks on code availability)

Version 3:

Reviewer comments:

Reviewer #2

(Remarks to the Author)

I thank the authors for addressing the comments that were previously raised, and have no further comments.

(Remarks on code availability)

Reviewer #1 (Remarks to the Author):

The manuscript by Gupta et al. uses the publicly available data on metagenome-assembled genomes with clinical isolates of *Klebsiella pneumoniae* and aims to find the genomic signatures of this species in carriage and disease. I have several major concerns regarding the analysis methodology and interpretation of results. Further, in my opinion, the manuscript lacks in advancing our understanding in the area. Some of my suggestions are provided below

We thank the reviewer for their feedback. We have revised the manuscript to address all of their technical concerns as detailed below. We would also like to emphasize that the main strength and advance of our work is in highlighting how the use of metagenome-assembled genomes (MAGs) can help us better understand the genomic differences between carriage and disease genomes, as clinical isolates are usually biased towards the latter. To our knowledge, this is the first large-scale integration of MAGs with clinical isolates to identify the genetic determinants of carriage vs disease in opportunistic pathogens. Using this novel approach we not only expanded the diversity of *K. pneumoniae* (doubled currently known gut-associated diversity), but also reveal biological functions (e.g., iron transcriptional repression and phage activity) that are implicated in *K. pneumoniae* pathogenicity. We believe these results will open new opportunities to further understand the pathogenesis of *K. pneumoniae* in their natural reservoir and encourage the exploration of similar patterns in other opportunistic pathogens.

1. Details on the selection of 662 genomes (319+343) are confusing. While the authors do explain the origin of MAGs, the selection criteria of isolate genomes used is completely missing.

We apologize for the lack of clarity regarding how the isolate genomes were selected. Both the MAGs and isolates included in our study were collected from the Unified Human Gastrointestinal Genome (UHGG) catalogue (Almeida et al. *Nature Biotechnology* 2021). The UHGG represents a comprehensive database of genomes from the gastrointestinal tract that were processed and quality-controlled in a standardized manner. The isolates available within the UHGG were those deposited in NCBI, PATRIC, IMG and two other culture collections (HBC and CGR) that were annotated as having been isolated from the human gastrointestinal tract. We have provided more details in the revised manuscript:

Lines 95-97: “The UHGG represents a comprehensive collection of isolates from two human gut culture collections^{19,20} and public repositories (IMG²¹, NCBI²² and PATRIC²³), alongside MAGs derived from >11,000 metagenomic samples worldwide.”

Lines 488-492: “Isolate genomes within the UHGG were originally retrieved by surveying the IMG²¹, NCBI²² and PATRIC²³ databases for genome sequences annotated as having been isolated from the human gastrointestinal tract. This set was complemented with bacterial genomes belonging to two human gut culture collections: the HBC¹⁹ and CGR²⁰.”

2. The lack of difference in virulence properties of carriage and disease genomes is reported. While authors also mention that disease-associated genomes in the gut can increase the risk of subsequent nosocomial infections. However, the lack of any difference in virulence properties in carriage vs. disease genomes suggests that infections from both groups are equally likely.

Although we agree in part with the reviewer, it is important to note that the virulence score calculated by Kleborate is only based on the detection of three genetic loci (*ybt*, *clb* and *iuc*).

Therefore, these represent only a small subset of the many potential genes/mechanisms involved in *K. pneumoniae* pathogenesis. Furthermore, as discussed in their original publication (Lam et al. *Nature Communications* 2021), these markers are not necessarily a direct prediction of clinical virulence. In fact, our GWAS analysis identified >400 genes associated with carriage or disease lineages, suggesting there are large genetic differences between these two groups. This included genes involved in the biosynthesis of the capsule polysaccharide, which has also been previously implicated in *K. pneumoniae* virulence.

Nevertheless, in the revised manuscript we have also added a new analysis of virulence and resistance scores among carriage and disease by specifically comparing carriage- with infection-associated genomes only (excluding other diseases). Interestingly this showed that both virulence and resistance scores were significantly higher when considering disease lineages strictly from infection. We have revised the text to include these new results.

Lines 208-212: “When considering all disease lineages (that is, sampled from individuals with infection or other diseases), virulence scores were not significantly different (Wilcoxon rank-sum test, $P = 0.516$) according to health status. However, comparing genomes from carriage with those only from infection revealed statistically significant differences ($P = 0.0028$; average score in carriage = 0.91, average score in infection = 1.18).”

3. Authors should mention the carriage/disease genomes ratio in MAGs and isolate groups.

We have now included this information in the revised manuscript:

Lines 99-101: “Of the 656 genomes, health status information was obtained for 521 genomes, with 132 being classified as carriage- (49 isolates, 83 MAGs) and 389 as disease-associated genomes (245 isolates, 144 MAGs).”

4. Was there any redundancy in the selected genomes? Did you use any dereplication method to cluster genomes and remove any redundancy?

We thank the reviewer for raising this point. We had not performed any dereplication on the UHGG dataset. However, we have now rerun all the analyses after dereplicating our collection using a Mash distance of 0 to remove duplicate or nearly identical genomes. This reduced the genome collection slightly from 662 to 656 genomes. All the values and results throughout the manuscript have been updated and the conclusions remain unchanged. We have added this additional step in the Methods as well.

Lines 504-508: “Lastly, all genomes underwent taxonomic verification for *K. pneumoniae* using Kleborate²⁴ v2.4.1 using default parameters, and subsequently dereplicated to remove duplicate or nearly identical genomes based on a Mash⁴⁸ distance of 0 with a sketch size of 10,000. The final filtered and dereplicated dataset comprised a total of 656 genomes (317 MAGs and 339 isolate genomes; Supplementary Table 1).”

5. Line 165-166: report p-values in all comparisons, even the non-significant ones.

We have now included a new supplementary table (Supplementary Table 2) with the effect size and P values of all PERMANOVA tests performed. It is worth noting that despite variations in effect size (R^2), all PERMANOVA tests were statistically significant ($P < 0.001$).

6. Line 204: needs some revision. Also, the results based on analysis of a single species should not be extended to the whole metagenomic community and host health states.

We agree with the reviewer and have removed this sentence. In addition, this whole paragraph (lines 201-225) has also been revised as described above in response to comment #2 and in response to the comments by reviewer #2.

7. Line 219 & 241: Please use consistent names for groups throughout the manuscript (disease/carriage, infection/carriage, carriage/disease state). This will improve the paper's readability.

We have now revised our terminology to make it as consistent as possible throughout the manuscript. However, it is worth bearing in mind that since we had genomes from both infection and other diseases, many of our analyses performed across all disease genomes were also confirmed using samples from infection only. Therefore, whenever we use the term "infection" we refer to disease-associated genomes specifically from infected patients (and not just with any disease).

Reviewer #2 (Remarks to the Author):

In the study by Gupta et al, the authors collate a collection of 319 metagenome-assembled genomes and 343 isolate whole genomes from gut carriage/disease to examine the genome diversity of carriage versus disease-associated Kp. My concern is that this does not represent a systematic collection of genomes, and there will likely be different sampling biases underlying the data collection/sequencing of MAGs versus isolate whole genome data, which will likely impact the observations/conclusions that the authors have made. For example, most studies looking at clinical infections select for AMR. Additionally, it appears that the majority of MAGs and isolates are not linked/associated with the same host/geography (i.e. there is a disproportionate number of health and/or disease associate genomes within different geographies) - is there any point then in trying to draw comparisons like ST trends and genome markers between carriage versus disease associated genomes when differences may be reflective of differences in the circulating clones/isolates within particular geographies?

We thank the reviewer for their comments and feedback. We fully acknowledge that large datasets such as ours that were collected from various sources inherently bring sampling/geographical biases depending on the original study design and the populations sampled.

To mitigate these biases as much as possible, we conducted several additional analyses. First, we repeated the ST analyses in two ways (lines 119-125): 1) by considering only genomes from countries where both carriage and disease genomes were available, and 2) by partitioning the data into isolates and MAGs. In both scenarios, the limited overlap between STs from carriage and disease genomes observed in the original analysis remained consistent (between 6–11%). Secondly, we replicated the virulence and antibiotic resistance analyses with isolates alone, confirming the observation of higher scores among disease-associated genomes (lines 220-225 and Extended Data Fig. 5). Lastly, we reproduced the GWAS using genomes only from countries that had both carriage and disease genomes available, while incorporating "country of origin" as a covariate in the GWAS linear model. This approach confirmed that 85% of the candidate genes identified in the original dataset remained significant (lines 253-257 and Supplementary Table 4).

While these steps do not completely eliminate the potential for sampling and geographical biases between carriage and disease genomes, they strengthen the validity of our conclusions, showing that these confounding factors are unlikely to be the primary drivers of our results. Nonetheless, we have also more explicitly discussed the above issues as some of our study's limitations in the Discussion (lines 315-323).

Other Comments:

- Line 100: the authors state here that genomes were classified as carriage or disease associated - was the metadata stated as such as were there various categories that were collapsed down into these two categories? If so, are they authors able to elaborate here on these definitions?

Yes, the reviewer is correct. There were various metadata categories available for each genome and we classified them into health and disease based on the following criteria: First, carriage strains were defined as those obtained from individuals explicitly classified as healthy in their original study. Individuals classified as diseased were further divided into two groups: (i) those with any disease, such as infections, colorectal cancer, autoimmune disorders, liver disease, among other conditions considered as risk factors for *K. pneumoniae* infection; and (ii) those

specifically with conditions directly associated with or caused by the colonization of pathogenic *K. pneumoniae* strains, namely infections and diarrhoeal diseases. This information has now been clarified in the revised manuscript (lines 514-520).

It is worth noting that all of the analyses of genomic signatures (virulence, antibiotic resistance, GWAS and machine learning) were performed using all disease genomes, as well as with only those from infection. This is explicitly stated in the revised manuscript throughout the relevant results sections.

- Lines 107-112: the authors state that there is more variability in the ST distribution of disease genomes versus carriage isolates but is this again due to differences in the sampling/geographies? For example, ST258 is a common carbapenem resistant Kp lineage in the US and ST512 in Italy, but there appears to be no genome data from 'healthy' patients from the US or Italy?

We would like to clarify that we in fact observed more diverse ST distribution in carriage, compared to disease-associated genomes. We believe these are not necessarily a reflection of geographical biases as the trends were consistent when subsetting the data to countries that had both carriage and disease genomes. We have also now calculated a Shannon diversity index for the different ST distributions to obtain a quantifiable measure of diversity. Values obtained using samples from all countries were: 4.45 (carriage) vs 4.08 (disease); values using samples from countries with both carriage and disease genomes: 4.21 (carriage) vs 3.37 (disease). We have now provided the specific values of diversity in the revised manuscript, as follows:

Lines 111-114: “Carriage strains on the other hand were more diverse (Shannon diversity = 4.45, compared to 4.08 among disease-associated genomes), with ST65, ST23, ST35 and ST29 being the most frequent, but only represented by a maximum of 6 genomes (Fig. 1b).”

Lines 523-525: “The diversity of STs among carriage and disease was calculated using the Shannon diversity index using the number of genomes assigned to each ST.”

- Lines 112-113: Can the authors elaborate on how many genomes from the disease versus carriage were these 20 STs represented? Any comments on country distribution? i.e. Are genomes with the same ST/carriage/disease being detected in the same country?

We thank the reviewer for this great suggestion. Among the 20 STs shared between carriage and disease, 83 were from diseased and 35 from healthy. We did identify three countries (Australia, China and Switzerland), where genomes with the same ST were classified as carriage and disease ($n = 43$). We have added this new information in the revised manuscript.

Lines 115-118: “These were represented by 83 genomes from disease and 35 from carriage. We also identified three countries (Australia, China and Switzerland), where genomes with the same ST were classified as carriage and disease (representing 43 genomes).”

- Lines 174-176: The authors state here that ST11/ST258 were absent from >11,000 faecal metagenomic samples; however the authors only looked at a subset of these samples (662 MAGs/isolates), so it's unclear how they can extend this observation (or lack thereof) to the entire sample collection? Additionally, will some of these 11,000 metagenome samples lack Kp - the authors mention $n=985$ Kp genomes from faecal samples in the methods section.

We apologize for the confusion regarding these numbers and terminology. First, we should clarify that MAGs refer to individual genomes, whereas faecal samples or faecal metagenomes denote the original stool samples containing a diverse microbiome community. The UHGG corresponds to a collection of >200,000 MAGs retrieved from >11,000 faecal metagenomic samples (many MAGs are retrieved from one sample). The original number of 985 genomes cited in our paper corresponds to all *K. pneumoniae* MAGs and isolates found in the UHGG before any filtering. After the filters with CheckM and GUNC this reduced to 656. This means that in the entire UHGG collection there are only 656 total genomes (317 high-quality MAGs) belonging to *K. pneumoniae*. Therefore, *K. pneumoniae* was not detected in the majority of faecal samples screened for the UHGG. Those that were positive for *K. pneumoniae* were included and analysed here. This suggests two things: (1) carriage of *K. pneumoniae* is rare (~3% prevalence, based on the number of high-quality MAGs retrieved) and (2) carriage of ST11/ST258 is even rarer (none detected among the UHGG MAGs).

- Lines 196-197: The authors should briefly clarify here what the resistance and virulence scores measure - I've just seen that they provide the description of this in the methods, but should still give a brief explanation in the results, particularly if the methods comes after the results section.

We have now added this information in the revised manuscript:

Lines 204-206: "This method calculates a virulence score based on three genotypic markers (*ybt*, *clb*, *iuc*), as well as a resistance score based on the presence of ESBL, carbapenemases and/or colistin resistance genes."

- Lines 197-199: Can the authors provide some quantification here between the differences in the scores for the carriage versus disease cohorts (i.e. the mean or median)?

We have now provided these measures in the revised manuscript:

Lines 210-215: "However, comparing genomes from carriage with those only from infection revealed statistically significant differences ($P = 0.0028$; average score in carriage = 0.91, average score in infection = 1.18). Furthermore, resistance scores were also significantly different ($P < 0.0001$) when contrasting carriage (average score = 0.05) to all disease lineages (average score = 1.11) or to those from infection only (average score = 1.76)."

- Lines 198-238: Are some of these observations driven by Kp clone characteristics? For example, the authors state resistance scores in the disease cohort are significantly higher but there is a higher proportion of MDR Kp (e.g. ST11, ST258, ST512) in the disease cohort, which carry higher loads of AMR.

Yes, we believe that specific *K. pneumoniae* clones are partially driving the resistance scores. Of the genomes with a resistance score >0, 68% belong to either ST11, ST258 or ST512. The remaining genomes are distributed across 16 other STs. We have added this information in the revised manuscript.

Lines 215-216: "Of the genomes with a resistance score >0, 68% belong to either ST11, ST258 or ST512, with the remaining genomes distributed across 16 other STs."

- Lines 446-449: Can the authors provide a little more explanation on the purpose for CheckM and GUNC? Is the >90% completeness only looking at core/chromosomal sequence?

We used CheckM and GUNC to quality filter the MAGs here included, as they are more likely to be incomplete and/or contaminated, compared to isolate genomes. CheckM uses a database of single-copy marker genes to estimate genome completeness and contamination. Therefore, it looks specifically at the composition of the core genome (which marker genes are present and whether they are found in more than one copy). However, GUNC investigates the full complement of genes to detect and quantify genome contamination (that is, the incorrect placement of genes from other species within a genome). It leverages an entropy-based measure of lineage homogeneity across contigs to determine if they share a consistent taxonomic assignment. Therefore, it is not specific to core genes and is able to detect evidence of contamination across the whole genome. We have added this additional information in the revised manuscript:

Lines 495-504: “First, we applied a filter of >90% completeness and <5% contamination based on the genome statistics obtained with CheckM⁴⁶ v1.0.11. CheckM uses a database of single-copy marker genes to estimate genome completeness and contamination, looking specifically at the composition of the core genome. Thereafter, we used GUNC⁴⁷ v1.0.3 to further exclude genomes with both a ‘clade_separation_score’ >0.45 and ‘contamination_portion’ >0.05. In contrast to CheckM, GUNC investigates the full complement of genes to detect and quantify genome contamination (that is, the incorrect placement of genes from other species within a genome). It leverages an entropy-based measure of lineage homogeneity across contigs to determine if they share a consistent taxonomic assignment.”

- Line 525: Can the authors comment on why this threshold was selected?

We chose a threshold of 0.005 distance (99.5% average nucleotide identity, ANI) for two reasons: (1) a previous study (Conrad et al. *Nature Communications* 2024) showed a pronounced intraspecies gap at 99.5% ANI among well-sampled bacterial species, suggesting the existence of defined intraspecies units within this ANI boundary; (2) we observed a clear bimodal distribution separated by a distance threshold of 0.005 when looking at the distribution of MAGs against their closest RefSeq genome (Extended Data Fig. 4). Even though the concept of “strain” is ill-defined, we believe both of these observations suggest that a 99.5% nucleotide identity is a reasonable threshold to define subspecific lineages in this population. We have revised the manuscript to better support the use of this threshold.

Lines 189-194: “Interestingly, the Mash distances exhibited a bimodal distribution separated between a distance threshold of 0.005 (approximately equivalent to 99.5% average nucleotide identity, ANI), potentially reflecting a strain-level boundary. In fact, a previous study showed a pronounced intraspecies gap at 99.5% ANI among well-sampled bacterial species, suggesting the existence of defined intraspecies units within this ANI boundary²⁷.”

- Figure 1: For panel B, it would be helpful to see a breakdown for each of the columns (i.e. ST counts) by MAGs versus isolate WGS

Thank you for this suggestion. We have added a new figure (Extended Data Fig. 1) that shows the distribution and overlap of STs after partitioning the data into MAGs and isolates. Even though the most prevalent STs between MAGs and isolates did not overlap, both comparisons

confirmed the limited ST sharing between carriage and disease genomes. This has been added in the revised manuscript:

Lines 119-123: “To mitigate potential geographical and sequencing biases, we further compared the level of ST overlap by only considering countries with genomes obtained from both carriage and disease (Fig. 1c), and after partitioning genomes into MAGs and isolates (Extended Data Fig. 1). These results further confirmed limited sharing of STs between health states (between 6% and 11%).”

- Figure 4: Similarly for all panels, it would be helpful to see which of the genomes correspond to MAGs versus isolate WGS, and whether this influences assembly of plasmids, which often carry the virulence loci and AMR genes that are captured by the virulence/resistance scoring of Kleborate

We thank the reviewer for this suggestion. We agree that given plasmids may be incorrectly placed in MAGs this could affect the estimates and comparisons of resistance/virulence scores. Therefore, we repeated these analyses using isolate genomes alone (Extended Data Fig. 5), which confirmed that virulence and resistance scores are higher among disease-associated genomes. This has been discussed in the revised manuscript.

Lines 220-225: “To account for biases related with the inability to correctly assemble plasmids in MAGs – which may carry the genetic markers used by Kleborate – we performed the same virulence/resistance comparisons using isolate genomes only. These results confirmed the above trends regarding higher virulence and antibiotic resistance scores among infection-associated genomes (Extended Data Fig. 5).”

Minor comments:

Line 36: Gram should have a capital G

Corrected.

Line 113 and 116: Typo, figure reference should be to Fig. 1C and not 2C

Corrected.

Lines 228-238: a lot of this text reads like discussion text rather than results

We agree and have moved part of this section to the Discussion (lines 299-310).

Line 255: do the authors mean >300 i.e. some of these 600 were whole genome sequenced from isolates?

Thank you for pointing this out. We have rephrased the sentence to clarify that 600 were both MAGs and isolates.

Line 265: unclear how this study would add to previous works looking at global diversity of Kp in animals as this study does not contain any sampling from animals

We agree and have rephrased the sentence to add another, more pertinent citation (lines 283-285).

Reviewer #1 (Remarks to the Author):

The work does not provide any new significant information, or methodology or any significant finding in the area of metagenomics or disease biology

Reviewer #1 (Remarks on code availability):

I just checked its availability but did not examine it.

We appreciate the reviewer's concern regarding the lack of novelty of our study. Given the comments by reviewer #2 we have expanded and entirely refocused our study to highlight the value of metagenomics in expanding our understanding of the diversity of *K. pneumoniae* in the human gut. Notably, we would like to highlight 3 key novel points:

- **MAGs represent a wide range of novel lineages.** At a genotypic level, 60% of sequence types (STs) assigned to MAGs were found to be new, and the majority were missing from clinical isolates (lines 105-122; Fig. 1). These findings highlight how metagenomics covers a much wider breadth of diversity of *K. pneumoniae* in the human gut. Whole-genome analysis also uncovered 86 MAGs with a genomic distance >0.05% (lines 205-212; Fig. 4) to any known isolate, providing a particularly larger expansion of lineages found in Singapore and the USA (lines 213-217; Fig. 4c).
- **MAGs harbour a unique set of genes missing from clinical isolates.** We identified and characterized new genes detected exclusively among MAGs. In fact, 214 genes found to be missing from all isolates were detected in up to 17 MAGs (lines 153-158), suggesting there are genetic features absent from isolates that are distributed in the human gut population (Fig. 2).
- **Discovery of new putative virulence factors.** Functional prediction and annotation of these novel genes using deep learning approaches revealed that half of MAG-exclusive genes may encode putative virulence factors, highlighting their potential clinical relevance (lines 161-168; Fig. 2). In addition, we identified one of these genes to have putative antimicrobial resistance activity (probability score 0.999), even though it did not match any known resistance gene based on sequence similarity alone.

We believe that building a more comprehensive genomic view of the global *K. pneumoniae* gut population has important implications for better understanding the asymptomatic carriage of this species; uncovering geographic and population-level gaps in existing isolate-based genomic resources; and bringing to light previously uncharacterized lineages and genes that may hold clinical relevance.

Reviewer #2 (Remarks to the Author):

The authors have addressed many of the reviewer comments, and the revised version is an improvement to the original manuscript. However, my concerns over the study's dataset remain - there are biases in the dataset which means any conclusions comparing diseased versus carriage will be flawed. As an example, the authors highlight less diversity amongst disease associated lineages and that these are dominated by ST11, ST258 and ST512; this is to be expected as it reflects the bias towards capturing prominent MDR clones that frequently cause outbreaks in clinical settings. The addition of MAGs does add value in expanding the availability of carriage associated genomes; are the authors able to reframe the narrative to focus on this aspect rather than disease versus carriage?

We greatly appreciate the reviewer's feedback and have now further taken their comments into consideration. In this revised version we have entirely reframed all sections throughout the manuscript (Title, Abstract, Introduction, Results and Discussion) to primarily focus on the value of metagenomics and MAGs to expand our understanding of *K. pneumoniae* gut diversity and reduce emphasis on disease/carriage comparisons. This included the following changes:

- Replaced the disease and carriage analysis of STs with the comparison of isolates versus MAGs (lines 105-122; Fig. 1 and Extended Data Fig. 1, see below). This showed that MAGs represent a much wider diversity of ST lineages, >60% of which were not found in isolate genomes.

Figure 1. Global collection of *K. pneumoniae* MAGs and isolates. *a*, Metadata distribution of the 656 *K. pneumoniae* genomes analysed. Data is partitioned into three metadata factors: genome type (metagenome-assembled genomes, MAGs or isolates), health status (diseased, healthy or unknown) and country of origin. *b*, Venn diagram showing the intersection between sequence types (STs) detected among MAGs (green) and isolates (blue) across all countries (top) or only considering countries where both MAGs and isolate genomes were recovered (bottom). *c*, Most prevalent STs of the gut-derived *K. pneumoniae* among MAGs (top) and isolates (bottom). *d*, Distribution of MAGs detected per country based on the number of locus variants (mutations) identified in the MLST genes in relation to a known ST profile.

Extended Data Figure 1. Genotyping of MAGs with low strain heterogeneity. *a*, Most prevalent STs of the gut-derived *K. pneumoniae* MAGs with an estimated strain heterogeneity <0.5%. *b*, Distribution of MAGs detected per country based on the number of locus variants (mutations) identified in the MLST genes in relation to a known ST profile. Only MAGs with strain heterogeneity <0.5% were considered.

- Removed the virulence/AMR score analysis between health and disease, and added a new section discussing the pan-genome patterns of MAGs (lines 153-168; Fig. 2). In particular, we investigated all genes found to be exclusive to MAGs above a 1% prevalence threshold (to account for possible contaminant genes). This revealed 214 genes missing from all isolate genomes, many of which potentially encoding virulence/AMR activity.

Figure 2. Distribution and functional annotation of MAG-exclusive genes. *a*, Probability scores obtained with DeepVF for each of the top 25 most prevalent genes exclusive to *K. pneumoniae* MAGs from the human gut. A threshold of 0.8 (horizontal dashed line) was used to classify putative virulence factors. *b*, Functional annotation based on Clustered Orthologous Groups (COG) for each of the top 25 most prevalent genes detected solely among MAGs. Genes are ordered based on prevalence, represented by the number of MAGs in which a gene was detected.

- Expanded the phylogenetic analysis of the novel lineages and quantified their contribution across different countries (lines 210-219; Fig. 4). This showed that Singapore and the USA benefited from the greatest phylogenetic expansion provided by MAGs (lines 213-219; Fig. 4c).

Figure 4. Comparison of metagenome-assembled genomes with reference isolate genomes. *a*, Distribution of genomic distances calculated with Mash between the *K. pneumoniae* MAGs and their closest reference genomes when compared against 20,792 *Klebsiella* isolate genomes from the NCBI RefSeq database. The histogram shows the frequency of MAGs at different Mash distance intervals, with a density curve (blue line) overlaid to illustrate the overall distribution. A threshold of 0.005 (vertical dashed line) was used to define lineages without a close reference genome. *b*, Metadata distribution of the 86 MAGs representing lineages without a reference isolate genome available on NCBI RefSeq. *c*, Phylogenetic diversity (PD) increase provided by the 86 MAGs without a matching isolate, compared to using isolates alone. Only countries where both MAGs and isolates were obtained were evaluated for this analysis. *d*, Phylogenetic tree of the 86 MAGs with a genomic distance >0.005 in relation to NCBI RefSeq, annotated based on their ST, country, source (community or hospital) and health status.

Ultimately, our key message is that by simply focusing on clinical isolates when studying pathogen evolution, diversity and function we are missing a substantial extent of *K. pneumoniae* diversity that is circulating in the human population. Metagenomics, specifically the use of metagenome-assembled genomes (MAGs) can help us mitigate this gap and open new opportunities to track the dissemination and evolution of lineages of clinical interest. In our view, these findings have important implications not only for the study of *K. pneumoniae* biology and function, but more broadly for understanding other opportunistic pathogens within the gut microbiome.

Despite the reframing of the manuscript, we would like to mention we decided to keep the mGWAS analysis of health and disease lineages for two main reasons. Firstly, in our mGWAS modelling we accounted for bacterial population structure, country of origin and genome type, which means that the

candidate genes we identified are more robust to ST/geographical biases. Secondly, the discovery of carriage genes involved in iron regulation could be very valuable for further mechanistic studies and provides a biologically plausible hypothesis that may be of general interest.

Reviewer #2 (Remarks to the Author):

In the manuscript by Gupta et al, the authors use human gut metagenome-assembled (MAGs) and isolate genomes to highlight the value of using MAGs at capturing additional diversity within the gut that otherwise would be missed using isolate whole genome data alone. Some of these conclusions have been derived based on their MAGs having ST locus variant calls; I can see two potential issues with this whereby the approach doesn't provide enough resolution to confidently distinguish two isolates as divergent (i.e. a single SNP within any one of the 7 genes can result in a locus variant being called, and secondly, have the authors considered that some of these locus variant calls can arise simply due to the impact of sequencing errors associated with shallow sequencing depth that can impact MAG accuracy?

We thank the reviewer for all their comments and suggestions that have helped us further improve our manuscript. To address their two main issues related with potential limitations of the ST locus variant (LV) analysis, we discuss below the following points:

- Resolution of ST locus variants. To address the concern raised about the limited resolution of ST LVs, we now show in the revised manuscript that the number of LV calls correlates with greater genomic differences. In short, we measured genomic distance by comparing each MAG to its closest match in the RefSeq database using Mash. Given that RefSeq contains over 20,000 *Klebsiella* genomes, it likely encompasses the majority of known STs, providing a reliable reference point for our analysis. We then compared genomic distances across MAGs with 0 LVs, 1–2 LVs, and more than 2 LVs (Extended Data Fig. 5, shown below). We found that genomes with more LVs had significantly higher genomic distances (Wilcoxon rank-sum test, adjusted $P < 0.05$). This means that a higher LV count reflects a higher genomic divergence, and not just isolated SNP changes.

Extended Data Figure 5. Comparison of ST variation with genomic distance. Comparison of the distribution of Mash genomic distances estimated for MAGs with no ST locus variants (0LV, $n = 120$), 1-2 LVs ($n = 160$) and >2LV ($n = 33$). Mash distances were inferred by comparing each MAG against their best-matching RefSeq genome. The centre line within the box represents the median score. Whiskers are shown extending to the furthest point within 1.5 times the IQR from the box. P values were derived from a two-sided Wilcoxon rank-sum test and corrected for multiple testing using the Bonferroni–Holm method.

- Sequencing depth and MAG quality. Regarding potential issues with sequencing depth and MAG quality, we would like to highlight two main points. First, we focused our analysis specifically on high-quality MAGs (>90% complete), which require much deeper sequencing than is typical in MAG-based studies that include MAGs as low as 50% complete. This substantially reduces the likelihood that sequencing errors or assembly artifacts contribute to the observed LVs. Secondly, as discussed in the manuscript (lines 183-191), we found a strong correlation between the diversity of core genome SNPs and differences in overall gene content (Mantel test, Pearson's $r = 0.77$, $P < 0.0001$). This pattern is inconsistent with sequencing errors, as artefactual SNPs would not be expected to systematically covary with gene content.

Taken together, these results indicate that the LV calls in our MAGs likely represent genuine biological differences rather than artifacts of low resolution or sequencing depth. Nonetheless, to address these concerns we have added these findings (lines 215–220) and a new figure (Extended Data Fig. 5) to the revised manuscript.

In the last results section, the authors look for genomic signatures associated with carriage versus disease across their dataset of MAGs and isolate genomes. Are the authors able to comment on whether the same results would have been attained had they used isolate genome data alone? This would be one way to demonstrate impact/added value of including MAGs for analysis.

We thank the reviewer for this suggestion. We repeated the GWAS with isolates only, showing that 70% of genes (237/339) previously reported overlapped, which we believe not only emphasizes the robustness of the observed signal, but also highlights how the use of MAGs was able to reveal additional signatures of health and disease. We have included this in the revised manuscript (lines 272-275).

In addition, the machine learning results we previously reported (Fig. 5d) showed that the performance of models combining both isolates and MAGs was significantly higher than using isolates alone (Wilcoxon rank-sum test, $P < 0.0001$). Collectively, we believe these results show that there is value in the use of MAGs for improving the identification of genomic signatures of carriage and disease.

Lastly, the authors mention that 25% of the MAGs were divergent but this means that 75% of the MAGs have a closely related RefSeq match (<0.005 mash distance); based on this perhaps the added value of adding MAGs to pathogen genomic studies is only incremental?

We appreciate the reviewer's observation. While it is true that ~75% of our MAGs have a close match in RefSeq (<0.005 Mash distance), we believe the remaining 25% still represent a non-negligible proportion given *K. pneumoniae* is one of the most extensively sequenced pathogens (the RefSeq collection contains over 20,000 sequenced genomes). Importantly, the RefSeq database comprises isolates from diverse sources (clinical, environmental, animal, etc.), whereas our dataset was exclusively collected from the human gut. This means that even though many of the genes and lineages may be detected elsewhere, this is to our knowledge the first instance they are being reported among *K. pneumoniae* gut genomes. Given that *K. pneumoniae* gut colonization is a risk factor to subsequent disease, we believe capturing a more complete picture of *K. pneumoniae* gut diversity will provide an important baseline to understand the future dissemination, transmission and evolutionary dynamics of the new lineages here discovered.

Other comments:

Line 26: missing among 'sequenced' clinical isolates

We have added this clarification.

Line 92: In the introduction, the authors emphasize the utility of MAGs for gleaning insights into colonising Kp/the microbiome; can the authors clarify here why they've also included isolate genomes?

We have now included a sentence to further clarify this.

Lines 96-98: “We combined both MAGs and isolates to understand the relative contribution of metagenomics in capturing the diversity of *K. pneumoniae* in the gut.”

Lines 109-110: Did any of the isolate genomes correspond to these STs?

Both ST29 and ST23 had isolate genomes in our collection, but none belonged to ST65. We have further clarified this in the manuscript.

Lines 109-111: “The most frequent STs detected in MAGs belonged to ST29, ST23 and ST65 (Fig. 1c), the latter of which was not represented by any isolate genome here included.”

Lines 113-115: Here the authors use ST locus variants as a measure of phylogenetic diversity and to highlight China/Fiji as therefore harbouring more distinct lineages; I don't think using ST variation can provide enough resolution to make this claim. A single SNP can result in a locus variant being called. Have the authors checked their phylogenetic tree to confirm that these genomes with >2 locus variant calls are elsewhere in the tree?

As mentioned above in response to the first reviewer's comment, we addressed this point by evaluating whether a higher number of LV calls corresponded to a higher genomic divergence. We found that genomic distances were significantly higher in genomes with >2 LV (Wilcoxon rank-sum test, adjusted $P < 0.05$) compared to those with ≤ 2 LV. We also observed that genomes with >2 LV had a median genomic distance >0.005 , again supporting that they represent more divergent lineages from their ancestor ST. We have added these results (lines 215-220 and 226-228) alongside a new figure (Extended Data Fig. 5) to the revised manuscript.

Line 119: typo, should be 'STs' instead of strains. Authors should probably clarify that strains with these STs generally have/or are typically associated with KPCs. Were there any MAGs that had these STs?

We have now corrected and clarified this in the revised manuscript (lines 119-121). In addition, we did not identify any MAGs belonging to ST11/ST258 isolates. As originally mentioned in the manuscript (lines 203-208 and Fig. 3), we believe this further suggests these KPC lineages may be very rare among the general population.

Line 126: can the authors clarify why they've used Panaroo with different configurations?

We decided to test various Panaroo configurations to assess how robust the results were to parameter choices. Given MAGs are expected to be of lower quality compared to isolate genomes, this could have

affected downstream pan-genome reconstructions. In the end, the results were generally consistent across different parameters (maximum variations in pan-genome size and number of core genes of 5% and 3%, respectively) so we chose the most conservative option. This has now been clarified in the revised manuscript (lines 128-130 and Extended Data Fig. 2).

Lines 154-157: I don't quite follow how the authors know that these genes are found exclusively in MAGs? i.e. which genomes/dataset have they compared their MAGs to?

We apologize for the confusion. In this section, the genes found exclusively among MAGs were defined based on the presence/absence patterns of the *K. pneumoniae* pan-genome reconstructed with our dataset. This means these genes were only found in *K. pneumoniae* gut MAGs and not in any of the *K. pneumoniae* gut isolates. We have clarified this in the revised manuscript.

Lines 158-160: "To further understand the unique genetic diversity of *K. pneumoniae* captured by metagenomics, we performed a dedicated analysis of genes found exclusively among MAGs within this genome collection (that is, missing from the *K. pneumoniae* gut isolate genomes)."

Lines 189: The authors state here that the MAGs corresponded to lineages that are not represented by clinical isolates; how do they know this? Have the authors included clinical isolates in their analysis or is this just based on earlier observation of MAGs that do not have defined STs/matching to those that have been defined in clinical isolates? As highlighted earlier, STs do not provide enough resolution.

We agree with the reviewer that this statement is too vague. We have now removed it from the revised manuscript.

Lines 197-198: Wouldn't their 317 MAGs represent 317 faecal metagenomic samples rather than >11000? Or do the authors mean that no *K. pneumoniae* MAGs were assembled from the other 11000 metagenomes?

Apologies for the confusion. Indeed, these 317 MAGs were the only high-quality MAGs assembled from screening >11,000 samples, meaning that no *K. pneumoniae* genomes were assembled from the remaining samples. This is clarified in the manuscript in lines 203-208.

Lines 212-219: The authors highlight here unique MAGs but aside from these being novel lineages that otherwise haven't been captured by whole genome sequencing, they don't elaborate on what additional value this brings? Also, the majority (75%) of their MAGs fall under the 0.005 Mash distance (i.e. are closely related to a RefSeq genome?); but the authors have not made any comments on this result.

As mentioned above, while it is true that ~75% of our MAGs have a close match in RefSeq (<0.005 Mash distance), we believe the remaining 25% still represent a non-negligible proportion given that the *K. pneumoniae* RefSeq collection contains over 20,000 sequenced genomes. Importantly, the RefSeq database comprises isolates from diverse sources (clinical, environmental, animal, etc.), whereas our dataset was exclusively collected from the human gut. This means that even though many of the genes and lineages can be detected elsewhere, this is the first instance they are being reported among *K. pneumoniae* gut genomes. We have discussed this in the revised manuscript:

Lines 231-233: “However, the remaining 231 MAGs did match a *Klebsiella* isolate genome from RefSeq, showing that these MAGs may be present in isolates from other body sites, hosts or environments outside the human gut.”

Given that *K. pneumoniae* gut colonization is a risk factor to subsequent invasive disease, having representative genomes from these novel gut-adapted lineages provides reference sequences that will facilitate future efforts to track their dissemination and evolution in both health and disease. We have now discussed this additional point in lines 238-242. Lastly, we would also like to highlight that the additional value of the use of MAGs is further illustrated by the ability to better discriminate health and disease populations (shown later in lines 283-288 and Fig. 5).

Lines 221-264: In this section,

Lines 282-283: This seems like quite a substantial number of unique genes associated with virulence/AMR - are these genes entirely absent from isolate whole genomes from NCBI etc?

As mentioned above, we would like to clarify that these MAG genes were originally identified as being exclusive to MAGs specifically within our dataset (i.e., missing from the *K. pneumoniae* gut isolate genomes). Based on the reviewer’s suggestion we have now also screened the entire *K. pneumoniae* RefSeq protein collection, defining a positive match when >80% of the query MAG sequence aligned with >90% amino acid identity against a RefSeq protein. This revealed that 36 of these MAG genes (17%) did not match known *K. pneumoniae* protein sequences. We have now added this additional information in the revised manuscript (lines 167-169 and 587-590).